# *Nibi* and Cultural Affordance at Walpole Island First Nations: Environmental Change and Mental Health

**DOI:** 10.3390/ijerph19148623

**Published:** 2022-07-15

**Authors:** Gerald Patrick McKinley, Regna Darnell, Dean Jacobs, Naomi Williams

**Affiliations:** 1Schulich Interfaculty Program in Public Health, Schulich School of Medicine and Dentistry, Western University, London, ON N6A 3K7, Canada; rdarnell@uwo.ca; 2Walpole Island First Nation, Wallaceburg, ON N8A 4K9, Canada; dean.jacobs@wifn.org (D.J.); naomi.williams@wifn.org (N.W.)

**Keywords:** first nations, environmental health, cultural affordances

## Abstract

Through an application of the Environmental Affordance (EA) Model, this paper explores the impact of environmental degradation on the community at Walpole Island First Nation. We outline how a change in relationships, broadly defined within an Anishinaabek ontology, can impact not only access to the local ecosystem but also how the affordances offered are lost. We base our argument on the local knowledge that *nibi* (water) is the system upon which all life depends and should therefore be the centre of efforts to maintain community-level mental health. Through the EA model we articulate the need to shift the focus from a human-centered ecosystem services model to an Anishinaabek relational ontology where relationships within ecosystems are bidirectional. As such, when those relationships are damaged through environmental degradation, local mental health and wellness are put at risk.

## 1. Introduction

*Nibi* (water) came before the Anishinaabek people, and, in parallel to the Anishinaabek creation story, the islands that make up what is currently known as Walpole Island First Nation (WIFN) emerged from the receding floodwaters at the end of the last ice age. The location of the Islands provided an abundance of foods, medicines, and knowledge to the ancestors of the current residents of the community and going forward. Foods include a variety of fish and wild game which reside in and near the islands. The convergence of two continental flyways and the abundant marshlands provide waterfowl as a staple in local diets. Medicines consist of sweetgrass and other plants that grow across the ecosystems of the community. However, it is understood by many Anishinaabek that a relationship with the medicines is necessary to ensure their effectiveness. The ancestors continue to provide guidance to their descendants spiritually and through the knowledge passed on through the generations which helps maintain the essential relationships. Communities learn to live a healthy life in relationship with their surrounding ecosystem (and the Beings within it) through the lessons passed on by their ancestors. These relationships are connected to nature through the cultural affordances offered by the ecosystem. However, if the relationships are impacted through environmental degradation, then mental health is vulnerable and can be negatively impacted because the environment is no longer able to offer the same health-maintaining relationships to the community. We define environmental degradation as changes to the natural environment caused by either the depletion of resources or the addition of damaging substances. Through an exploration of evidence from WIFN, we argue that the impacts of change to the environment can be seen in the existence of chronic stress levels of community members at WIFN. The chronic stress, in turn, has a negative impact on community-level mental health and vulnerability, which fuels ecological grief, worry, and depression.

Our understanding of who the Anishinaabek people are must be separated from contemporary, modernist concepts of nations. The Anishinaabek are the relations, language, and knowledge that are held by the descendants of the people who have inhabited the areas around the Great Lakes region since time immemorial. There is great diversity within and among the many communities which consider themselves to be part of the Anishinaabek Nation. Their language, Anishinaabemwin, is part of the Algonquin language family which ranges from the East Coast of North American to the Rocky Mountains. The Three Fires Confederacy of the Anishinaabek consists of Ojibwa, Odawa, and Potawatomi Nations. While each community is connected as Anishinaabek through shared language, knowledge, traditions, and values, they also maintain a high degree of local diversity which is reflected in their local relationships with the ecosystems they live with. WIFN includes members of the Three Fires Confederacy and descendants of the Shawnee who fought with Tecumseh during the War of 1812. 

Together, we outline an argument which is based on how the lack of previously present protective factors can impact the mental wellness of a community. Our focus is not to provide novel data on the current levels of pollutants in the *nibi* surrounding the WIFN. Our focus is on the social impact of those pollutants when they contribute to higher levels of chronic stress. In this paper we interlock three approaches. The first is an application of the Environmental Affordances (EA) Model developed to understand health inequalities. We use the EA model, specifically the seven pathways of this model, to allow us to shift of focus away from individual mental health to community mental wellness by taking into consideration the ongoing impacts of colonization. In this context, colonization is seen as a shift to a global-capitalist system which uses the traditional territories of the local Anishinaabek as a dumping ground for environmental pollutants. Additionally, this model allows us to consider relationships, and changes to them, in a systematic fashion. Second is a shifting of the focus towards an Anishinaabek perspective on wellness. Third, we use an ecosystem services approach to articulate the role of cultural ecosystem services to understand how the community of WIFN define the value of the gifts (affordances) provided by the ecosystems they live in. We argue that a systematic approach allows us to piece together the many threads of research that have been and continue to be conducted with the WIFN. Where answers are not clear, we recommend future research questions. 

Our objective for this paper is to explore how changes to larger relationship networks with community-level mental wellness. We argue that the relationships which connect the community with the ecosystems in which they live and the Beings who reside within supported the development of a health community, we hold that *nibi* is the driver of these relationships, so that a change to *nibi* can lead to a change in the relationships and to the community’s mental wellness. We take a systems approach in this paper. However, we work to establish that *nibi* is the system that matters and the mechanism through which the system works is relationships. Essentially, *nibi* forms and informs the local ecosystem within which the WIFN exists. That ecosystem is a connection to a larger regional, continental, and global set of systems [1]. The ancestors of the current community learned how to live *biimaadiziwin* (roughly, the verb for “a good life”) in the local ecosystem. They established relationships with *nibi*, other local Beings, which includes Beings that swim, crawl, and fly, as well as those that grow on the land and water. We believe that the process of culture takes place within the system which afforded everything that the ancestors needed to live. We focus on affordances because they are descriptive and prescriptive [2]. They are descriptive because they explain how local environment was and is used for the benefit of the community [2]. Additionally, they are prescriptive because they dictate culturally normative behaviour within the environment and the related community [2]. Changes to affordances results in changes in relationships through alterations in use patterns and the resulting social relationships and the conditions that support mental wellness. Environmental degradation increases chronic stress through these changes as well as through stress over the impacts of the pollutants themselves.

## 2. Environmental Affordances Model

The Environmental Affordances Model (EA) (Figure 1) is an integrative framework developed to understand the origins of mental and physical health disparities in communities [3]. The model itself is helpful specifically because it focuses on a means to explore the pathways that exist between the social and physical environmental contexts of the community, self-regulation behaviours, and how environmental affordances may help or hinder the development of good health. The EA model is a shift away from the individual towards community-level systems. Importantly, the EA model focuses on identity as a process of experiences rather than a fixed racialized “thing” and incorporates three approaches to understanding health outcomes resulting from those experiences: the fundamental causes model, the stress process framework, and the socio-biological perspective [3].

The fundamental causes model (FCM) was developed by Bruce G. Link and Jo C. Phelan [4] to explain how socioeconomic status (SES) and the resources it provides (money, knowledge, prestige, power, and social conditions) protect or hinder health outcomes. In short, the fundamental causes model directs our attention towards social inequities and the impact that they can have on the health of a community. A stress process (SP) framework identifies the interactions between experienced stressors and “social and personal resources that may have direct effects and/or mediate or moderate the health consequences of stress exposure [5]”. Importantly, the SP framework responds to chronic stress and acknowledges the cumulative impact it has on our health. Specifically, chronic stress impacts our bodies through a process of physiological dysregulations that takes place when regular functioning (homeostasis) is impacted, and the body releases increased levels of stress hormones [6]. Finally, the socio-biological perspective focuses our attention on the identification of specific pathways that may link environmental factors to health outcomes [3]. This approach asks how the social environment impacts the physical body. 

In this paper we explore the seven distinct pathways outlined in the EA model. Path 1 explores the local exposure to chronic stressors and how stress is patterned by social structures and contexts within the context of the Island and the neighbouring “Chemical Valley”. Path 2 ties the context to coping behaviours that are available to the WIFN. Here we explore changes to food consumption patterns and the associated social impact to a transition from hunting and fishing to store bought foods. Paths 3 and 4 explore the evidence for chronic stress and increased risk of stress related psychopathology at the WIFN alongside the literature and evidence connecting chronic stress to increased risk of morbidity and mortality. Path 5 discusses the literature on the relationship between chronic stress and poor health behaviours within the context of the WIFN. Finally, paths 6 and 7 discuss how self-regulatory health behaviours can increase the risk of medical morbidity and mortality beside the potential benefits of positive self-regulatory health behaviours. 

## 3. Anishinaabek Traditional Knowledge and Water

We frame our application of the EA model within local Anishinaabek knowledge and values. We argue that when we consider environmental affordances there is a subjective element to the local understanding of what is provided. For example, the catching of a muskie (or *maashkinoozhe*) can mean food, a trophy, a maintained relationship with the river and the fish inhabiting it, or confirmation of the health of the fish stock, depending on who catches it. In the section which follows, we articulate specifically how water plays an important role in the life of the community on the Island and how water is tied to the concept of environmental affordances through a relational ontology. In doing so, we reframe the system within which we are assessing the health impacts associated with environmental damage to one where relationships with *nibi* are the system rather than the imposed economic systems brought by global-capitalist/colonial powers. By shifting our focus to *nibi* as the system we are able to embed our arguments within an Anishinaabek relational ontology rather than using an external system and authorities as our reference point.

Based on Bruno Latour’s reminder that the “social” is born when two actors engage, we argue that all societies, which consist of more than two beings, have a value system that is based on the concept of relationships [7]. It is in how members of the society view those relationships that values become actions. Take, for example, the dominant Western–Euro–Canadian values on our relationship with water and the environment. Here, the relationship is hierarchical in nature, with water and the environment subsumed under the concept of “natural resources”. Natural resources are things to be controlled, administered, and incorporated into the dominant economic system for sale within markets. In this set of relationships, the environment is a set of commodities over which “man” has dominion. This perspective governs Canadian property laws, mining laws, and circumscribes our environmental policies by assuming that the environment serves economic growth [8].

Anishinaabek values are based on relationships and responsibilities that extend beyond the dominant Western–Euro–Canadian concepts of the environment and natural resources [9]. As Anishinaabek scholar, Aimée Craft [10] states “Anishinaabe *nibi inaakonigewin* (water law) tells us that water is life—*nibi onje biimaadiiziiwin*. We are born of water, and we are primarily composed of water. Not only does it give and take life, it is also a living being in and of itself that relies on a larger web of relationships to be well and to bring wellness to other beings”. In Anishinaabek culture, women are given the traditional responsibility to protect and care for the water, while the respect for the water is shared individually and collectively. Because we are all interconnected, what one does to *nibi* is viewed to impact all, and the protection we aim to provide is for all life, as *nibi* is life. *Nibi* is life and has life and all living things are dependent upon it. Put another way, *nibi* is not part of the economic system, as life and having life, *nibi* is the system. Responsibilities tie us to our relations which include ancestors, the Spirit world, other-than-human Beings, all peoples, the local community, your clan, and those yet to be born [9]. The word *biimaadiziwin* is often used as a noun to represent health. However, like much of *Anishinaabemowin,* it is a verb. *Biimaadiziwin* is about living a good life. But this raises the questions of “what is a good life?” Good for whom? Is the acquisition of good and continuous growth “good”? Local knowledge tells us that the focus on a good life is not about being good as an individual. *Biimaadiziwin* ties Anishinaabek people to a responsibility to live a good life, one that is lived through a responsibility to relationships. Deborah McGregor [9] notes that it is not enough to know something. What matters is our conduct or “the way we behave, the way we act upon our knowledge, the way we conduct ourselves in our relationships and responsibilities”.

Everyone has responsibilities, but not everyone has the same responsibilities. Anishinaabek teachings explain how responsibilities differ over the life course, but it encompasses the whole community. Traditionally children were cared for by the whole community. Movement was in family groups that included multiple generations. Children had a responsibility to learn how to act from their families. As the child grew, they began their “fast life” where they begin the rapid transformation into a young adult [11]. Elders begin playing a more important role in the lives of youth in this phase, teaching them more complex knowledge and introducing their fast prior to puberty. The “wandering life”, from 13 to 18 years old, is a time of increased responsibility for your own actions and it sets up the adult years [11]. Being an adult means accepting responsibilities to children. Finally, the Elders are responsible for teaching others through their actions how to be a good person. At WIFN, this is present in the environmental philosophy of Niin.da.waab.jig, the locally run research center. Here, the environment is something upon which the community has always depended on and will continue to be responsible for protecting for future generations [12]. The responsibility to the environment carries with it a knowledge that the environment is a source of health and livelihood for members of the community [13].

Contrary to ecological perspectives on culture, where culture is a cognitive process of understanding the environment in which one lives, for the Anishinaabek at WIFN, culture is given as a set of “original instructions” through which their responsibilities to their relations are laid out. While performance of their responsibilities may change as technology changes, the overall set of responsibilities remains intact. 

## 4. Ecosystem Services Models

We further our application of the EA model through an integration of an ecosystem services model. However, this too must be adapted to align with local knowledge. As identified by Sandifer and Sutton-Grier [14], the concept of ecosystem services is explicitly human-centered. Ecosystem services are the benefits that humans gain from the natural environment and are generally divided into four main areas: provisioning services such as food, water, timber, and fiber; regulating services that affect climate, floods, disease, wastes, and water quality; cultural services that provide recreational, aesthetic, and spiritual benefits; and supporting services such as soil formation, photosynthesis, and nutrient cycling [15]. The health of the ecosystem matters as higher quality ecosystem provides more options for health than a stressed or degraded ecosystem which may cause negative psychological and physical health impacts [16]. For the purpose of this paper, we are largely focusing on cultural ecosystem services (CES) which “reflect peoples’ physical and cognitive interactions with nature and are increasingly recognised for providing non-material benefits to human societies” [17]. CES are perhaps the least researched of the categories of ecosystem services [16]. However, we argue that the affordances that CES offer to a group of individuals are what forms them into a community.

When we discuss CES, we are talking about the ways in which people or communities engage with their local ecosystems. We argue that CEC are also akin to cultural affordances, where affordances is defined as what an environment offers [18]. As such, cultural affordances are services that Beings engage with [2]. Ramsteand, Veissiere, and Kirmayer [2] divide their definition of cultural affordances into natural and conventional affordances. For them, natural affordances are “possibilities for action, the engagement with which depends on an organism or agent exploiting or leveraging reliable correlations in its environment with its set of abilities”. For example, a person’s ability to coordinate their fingers and thumbs allows them the agility to fish, thus affording food. Conventional affordances are “possibilities for action, the engagement with which depends on agents’ skillfully leveraging explicit or implicit expectations, norms, conventions, and cooperative social practices [2]”. Here, affordances become more dependent on social and cultural factors. Conventions require a knowledge of local norms within the community, which requires the ability to communicate with others and a means to store that knowledge. 

In Anishinaabek culture this is often achieved through the telling of stories, including the very old *aadizookaan* through to contemporary tales of community experiences. A story contains information on how to live *biimaadiziwin* and how to have good relationships with all the Beings around you. For the community at WIFN, conventional affordances need to include very real relationships with the environment where the symbolism of the convention must be tied to acting your responsibilities. Through inter-connecting natural and conventional cultural affordances, we begin to understand how others expect us to act and behave within a specific social context. Arguably, we adapt our conventional cultural affordances from our natural ones in the process of learning how to act together, passed down from our ancestors. 

## 5. Path One

We begin our application of the EA model by first exploring how the local context at WIFN may increase exposure to chronic social stress and how the exposure may be patterned by social structures. Next, we discuss how stressors may impact the community over the life course and compare how differential exposure to chronic stress creates and perpetuates differences between members of the WIFN and neighboring non-indigenous communities. We do this by detailing the physical environment and discuss how the affordances offered by the environment contributed to the health and wellbeing of the community. Next, we discuss the current and ongoing environmental stressors which are impacting the community and their associations with colonization and globalization. 

The WIFN is located on unseeded territories where the St. Clair River meets Lake St. Clair and is at the heart of the Great Lakes. The name Bkejwanong, which means “where the waters divide” in Anishinaabemowin, reflects the geographical location of the community. *Nibi* encompasses the community, it is present in all aspects of life in the community. The WIFN occupies Walpole Island, Squirrel Island, St. Anne Island, Seaway Island, Bassett Island, and Potawatomi Island. Through the actions of the surrounding *nibi*, the community has a relationship with five ecosystems on the islands: coastal waters, wetlands, tall grass prairie, oak savanna, and forest. Wetlands on the island account for approximately 6880 hectares or 17,000 acres. The tall grass prairies, wetlands, and oak savannas are one the largest remnants of these ecosystems in Ontario which are important elements of WIFN’s territories, are among the most valuable forms of ecosystems [19]. Historically for the ancestors of the current community at WIFN, the waters provided food in the forms of fish, waterfowl, amphibians, and marine mammals. The land provided food from larger mammals and local flora as well as materials for housing and plants for medicines. Local Anishinaabek culture grows from their relationship with these ecosystems. Many of the traditional skills and knowledges are still held and practiced by community members. 

The *nibi,* which provided the islands and all that they afford the community, has also contributed risks when we do not act responsibly. The wetlands and sediments that make up the islands allow for a significant sink effect where the sediments can hold a variety of contaminants that can contribute to increasing the risk of toxicity [20]. The source of the contamination is well known locally. The community has been forced to live with ongoing chemical spills from petro-chemical facilities in Sarnia’s “Chemical Valley”, which is located upstream on the St. Clair River. While not the only contributor to the pollution, Sarnia is home to 40% of Canada’s petro-chemical industries, and the St. Clair River was designated an Area of Concern in 1987. Recently, 34 of 35 audited plants were found to be in violation of one or more environmental protection regulation [21]. Additionally, a number of tributaries and drains connect to the river bringing effluents from sewage treatment plants, runoff from agriculture, and other industrial contaminants from both the Canadian and American sides of the river towards the WIFN [22]. Two major persistent pollutants of concern in the community continue to be the mercury (Hg) and arsenic (As) in the sediments and *nibi* [23]. 

Chemical Valley is the name given to the collection of petrochemical plants which make up an industrial corridor along the Canadian side of the St. Clair River. Chemical Valley processes approximately 40 percent of Canada’s petrochemicals [24]. The plants are examples of the impact of poor regulation of environmental pollution in Canada, with Canadian plants producing more pollution than comparable facilities in the United States [25]. The pollutants originating from Chemical Valley have impacted the air quality, water quality, soil, fauna, flora, and human populations living in and around WIFN [21]. The chemical plants produce large quantities of particulate matter (PM) which are released into the air and contain heavy metals in trace quantities, which can potentially have adverse effects on humans and environmental health [26]. Muttray et al. [27] identified that, while overall pollutant rates have decreased over the past 12 years, every sampled fish species from the St. Clair River “exceeded the Canadian tissue residue guideline for PCBs for the protection of mammalian wildlife consumers of aquatic biota”.

The effect of Chemical Valley on the stress of the population at WIFN and neighbouring First Nations is well documented [28]. Specifically, Henley et al. [29] confirmed increased hair cortisol concentrations in a sample of WIFN volunteers compared with volunteers from a local non-First Nation community. These results are associated with higher levels of chronic stress in the community tied to, among other things, health risks linked to environmental degradation. We argue that the mechanism through which this stress is produced is the impact that the environmental contamination has on the relationship that members of the WIFN community can have with their ecosystem, particularly those Beings who live in or near *nibi*. 

## 6. Path Two

In their work defining the EA model, Mezuk et al. [3] contrast a lack of access to healthy foods and recreational physical activity with greater access to tobacco and alcohol retailers, corner stores, and regulated drugs. In turn, this leads to increased consumption of high sugar, high fat, processed food, and the use of tobacco and alcohol as a means of responding to chronic stress. In the context of WIFN, we argue that Path Two has a social aspect as well as a consumption aspect. Our argument is based on the Anishinaabek knowledge that they are spiritual Beings living a physical existence. Therefore, we must consider the impact that changes in relationship between the community and their relations in the greater ecosystem has and its impact on their spiritual Self. Importantly, this makes the impact of increased environmental degradation about more than access to food. It is about the removal of protective factors that contribute to the whole Self. Not being able to fully or safely access the process of gathering food or medicines, even if it is a perceived lack of access, can negatively impact the mental health of community members. Additionally, individuals may become disconnected from the foods by avoiding eating traditional foods as a way of preserving the food itself, especially during times when we are taught it is not right to do so or it is under environmental threat. Overall, the changes decrease the use of protective factors. 

From a holistic perspective, being consists of a Spiritual Self, Mental Self, Emotional Self, and Physical Self. The separation of one from the whole is impossible. Similarly, the affordances offered by an action “feeds” all four aspects of the Self. For example, fishing provides food for the physical Self and decreases dependence on store bought, processed foods. The emotional Self is provided with a purpose. The emotional Self is regulated through the process and activity of fishing itself. Finally, the spiritual Self is supported through the process of connecting with nature, from being in relationship with *nibi*, and from using traditional knowledge of the ecosystem. The intergenerational aspect of teaching a relative to fish, prepare the fish, and the process of eating together improves mental health through increased access to an individual’s social support network. Interestingly, higher levels of social support were “significantly related to a lower likelihood of depression or anxiety for females” in First Nations communities [30]. Perceived and actualized social supports are believed to buffer us against stress. Additionally, recent research connects social support with decreased neural threat responses and decreased risk of the associated mental disorders associated with stress [31]. As such, social supports are actualized through a performance of activities with our relations and when those actions are not available because the affordances offered by the ecosystem are at risk, then mental health can be at risk.

In terms of mental health, Anishinaabek knowledge helps us rethink ecosystem services as more than what is given to the human population by allowing us to consider *biimaadiziwin* as the process of living out one’s responsibilities to Creation. To live a good life, communities need to have ongoing, bidirectional relationships with the natural environment around them and the Beings who dwell in that environment. The research we conducted to prepare our report of Health Canada provided additional evidence that community members continue to change their behaviours towards consuming local foods over worries about the impact of environmental degradation [23]. The 24-hour dietary recall survey we completed as part of this work indicated that most of the participants are eating store bought food rather than local game. 

## 7. Path Three

We know, based on an abundance of research, that chronic stress is bad for you. Chronic stress leads to the activation of the endogenous stress systems, specifically, the hypothalamic–pituitary–adrenal (HPA) axis [32]. Repeated activation of the HPA-axis is associated an increased sensitization of the system and an increased risk of depression [33]. Alarmingly, sensitization means that through repeated exposure to stress we become sensitized to stress so that even minor stressors can become increasingly triggering of depressive episodes [34]. 

Stress within the WIFN has repeatedly been connected to worry over the health of children and grandchildren in the community [35]. Additionally, the WIFN have not been immune from the impacts of historical and ongoing colonial actions in Canada. Joe Gone articulates that this form of socio-psychological stress is “complex, collective, cumulative, and intergenerational” rather than individual [36]. As such, it impacts whole communities regardless of their direct exposure to the initial stressor. Importantly, the environmental stressors are a result of actions by colonial governments and non-indigenous owned, often multinational, businesses who transfer the benefits of their actions away from the source of the environmental risks. As a result of both the environmental and colonial stresses, the conditions which promote an ongoing cycle of stress and stress sensitization are present across the life course for residents of the WIFN. 

## 8. Path Four

Mezuk et al. identified social adversity being associated with higher allostatic load, stressful life events associated with elevated risk of chronic health conditions, and perceived discrimination being associated with increased mortality in their discussion of Path Four of the EA model [3]. However, as mentioned above, we are seeking a wholistic understanding of the impacts of environmental stress on mental health. As such, we expand beyond the allostatic load approach to consider an impact on the community’s ability for *biimadiziwin* though a decrease of engagement with *nibi* and the larger community within their ecosystem. Cited in Beckford et al., one Elder expressed this loss as “we are totally dependent on our land. Without the river we would have no fish, without the marshes we would have no ducks, without the *mishkodi* we would have no medicines, without the beauty of nature we would have no peace. The land is our soul [22]”. In short, we cannot discuss the mental health of the community without a wholistic understanding of their relationships and the whole Self. 

Community members are aware of the impact on their health associated with the contamination of the river and surrounding ecosystem. Community members have collected evidence showing that industrial pollution from Chemical Valley continues to contaminate fish and other marine life [23]. Through a process of bioaccumulation, contaminants in the sediments are consumed first by smaller organisms before being eaten themselves by larger fish and wildlife. Contaminated vegetation is consumed by herbivore and omnivore species before residents consume the local foods. Many residents are afraid to eat fish from the river or drink water from the public system in the community because of the perceived health risks. 

Historically, the health concerns at WIFN have focused on diabetes, cancer, and other physical health concerns. Research from across Canada has connected these health outcomes to a history of ongoing colonization [37]. However, recently mental health has become a primary concern driven by a sharp increase in substance misuse challenges. The community declared a state of emergency over substance misuse and deaths in the summer of 2021 [38]. There is strong evidence for a connection between chronic stress and substance misuse [39]. Specifically, the effects of chronic stressors, including trauma, on “psychological distress is partially mediated by substance misuse”, and the cumulative exposure to stressful events “drives a feedback loop between substance misuse and psychological distress [40]”. Here, we argue that there is a need for greater research to explore the mechanisms through which perception of environmental stresses manifest themselves in stress-mediating actions such as the full spectrum of substance misuse. Specifically, we would like to understand cultural and biological mechanisms which connect perception through action so that we have a better way of understanding why risk causes actions. 

## 9. Path Five

For Mezuk et al., Path Five is based on three main points: stress increases food intake while simultaneously providing less pleasure from that food and increasing desire for comfort foods; stress is associated with behaviours such as smoking and alcohol and substance misuse; and stress is associated with overall poor health behaviours [3]. In the context of the WIFN, we add that stress and lack of confidence in safety associated with environmental degradation leads community members away from health-protective activities, specifically, the application of traditional skills in hunting and fishing. The protective potential of traditional skills has been documented by David Dunto, Russ Walsh, and Jocelyn Sommerfeld in northern First Nations communities. They note that “programs included the transfer of traditional skills and knowledge related to living on the land, and were largely taught through demonstration and collaborative work, which enhanced participants’ sense of identity and emphasized bringing together youth and Elders to foster intergenerational connection [41]”. As such, the lack of active healthy protecting behaviours should be considered a key aspect of the increased risk to mental health within the community and, based on known pathways listed above, is associated with an increase in substance misuse and other poor health behaviours. We identify this as an important area for future research. Presently, little is known about how individuals within the community are responding to chronic stresses associated with environmental degradation. 

## 10. Path Six

The increase in negative mental and physical health outcomes associated with the ongoing environmental degradation in the territories of the WIFN are not necessary, they are negative externalities caused by the decisions which led to a cluster of petrochemical plants on this stretch of river. They are the direct results of human activity and the dependence of our world on the petrochemical industry. Path Six deals with self-regulatory behaviour. However, those behaviours must be placed in the context of the greatest risk coming from decisions that are made outside of the community. There is abundant evidence to confirm that smoking, consuming alcohol, or engaging in other substance misuse is bad for your health. However, the focus tends to be on individual actions. From a public health perspective, we see a need for increased research led by local First Nations into the fundamental causes of those behaviours within the context of the WIFN. 

## 11. Path Seven

Mezuk et al. cite a number of important studies which confirm that smoking, eating palatable foods, or even substance use can dampen the body’s response to stress [3]. In their work they aspire to understand a paradox in the health outcomes of African Americans in the United States. In that context, African Americans “have higher risk of medical morbidity relative to non-Hispanic whites, (but) blacks have lower rates of common stress-related forms of psychopathology such as major depression and anxiety disorders [3]”. However, the evidence from Canada indicates that First Nations individuals have both higher rates of negative physical and mental health outcomes relative to the non-First Nations population [42]. This difference suggests that there may be another mechanism which needs to be identified which acts as a protective factor in African American populations may be missing in First Nations communities. Alternatively, there are contextual issues that are in need of further research within First Nations communities. Our argument is that the negative impact that environmental degradation has had on the community’s ability to maintain its relationships within their local ecosystem has overcome any potential short-term impacts from diet, smoking, or substance misuse.

## 12. Conclusions

We identify several benefits to integrating the EA model, an Anishinaabek perspective on wellness, and an ecosystem services approach. The first is a health equities approach to understanding mental health outcomes. This is an important transition away from individual approaches to mental illness which locate the problem in single individuals. The health equities approach provides a rebuttal to a lingering, racist belief that First Nations people are somehow flawed and that these flaws are the cause of their poor health outcomes. We also see an important step in decolonizing discourse that is associated with indigenous health through an shifting of that discourse to a local First Nation’s perspective rather than using local knowledge as an secondary form of knowing. We have worked to centre this paper on *Nibi* and relationships rather than economic gain and resulting degradation. Finally, we tie our argument to our understanding that local ecosystems shape communities through the process of ancestors learning how to live a good life in relationship with what is provided. From these teachings a community learns how to be healthy and when the relationships are damaged there is a greater risk of the community experiencing harms. 

While the EA model was developed to support research into the health outcomes in African American populations, we believe that it offers potential as an approach for understanding the relationship between environmental health and the mental health of a First Nation community. Specifically, the systematic approach to the stress and response system allows us to view the health outcomes at WIFN at the community level, rather than taking a clinical perspective with focus on the individual. There are great benefits to public health research in an affordances approach to relationships. To draw on an analogy from language studies, there is a significant difference between what a word denotes compared to its connotation. By this we mean that culture and traditional knowledge play an important role in shaping what something provides to our mental health based on our perception of that phenomena. 

The prescriptive aspects of conventional affordances likely shape individual behaviours within a collective community. As such, culture and traditional knowledge may play an important role in keeping a community healthy through the social regulation of relationships. However, if those relationships, broadly speaking, are negatively impacted by environmental degradation then there is a greater risk of increased chronic stress through a loss of culturally normative social relationships. The loss of these relationships may explain some of the differences in health outcomes between African American populations and First Nations communities when both are facing chronic stressors of different forms but potentially similar degrees. 

Returning to the statement that *nibi* is the system within which all life exists, it is through this system of relationships that the protective factors that kept the ancestors of the WIFN in good mental health by providing the means to live *biimaadiziwin* were provided. The relationships that *nibi* provides are central to health. Unfortunately, colonization and the imposition of a global economic system dependent on resource extraction has negatively impacted how the WIFN is able to interact with their ecosystem and the affordances offered through those relationships. The mechanism through which stress can be caused can just as easily be the absence of protection as the presence of risk. For example, neglect of a child (the lack of a positive parental relationship) can be as strong an adverse childhood experience as physical violence. 

Joe Gone notes that indigenous knowledge understands the importance of ‘‘culture as treatment’’ as a potential response to the chronic stresses associated with the environmental degradation that is present at WIFN [36]. We believe that a return to traditional relationships built on an understanding of the greater responsibilities each person holds provides a valuable approach to responding to chronic stress at WIFN. While we have applied the EA model to organize our argument and identify potential new areas for research, we believe that all approaches to supporting the mental wellness of the community must be translated to incorporate local traditional knowledge systems so that academic colonialism does not undermine local control of their lives. Moving forward, we will engage in greater relationship-focused research within the community to understand how members are interacting with each other and with their relations in the larger ecosystem.

## Figures and Tables

**Figure 1 ijerph-19-08623-f001:**
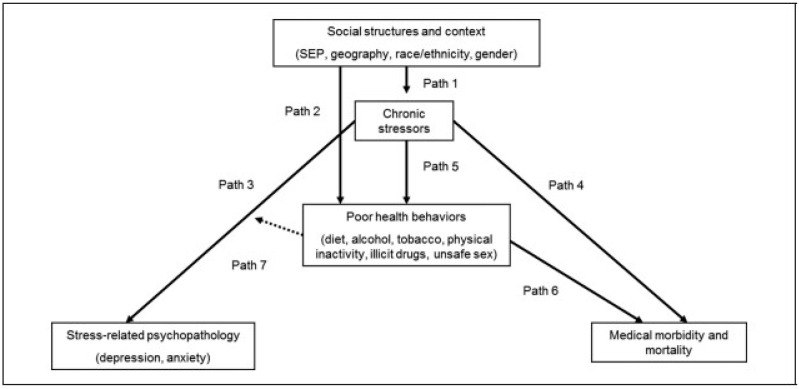
Source: The Environmental Affordance Model. Reprinted with permission Ref. [3].

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
