# Peer review of "Nibi and Cultural Affordance at Walpole Island First Nations: Environmental Change and Mental Health"

_ijerph, 2022, doi:10.3390/ijerph19148623_

Round 1
Reviewer 1 Report
Dear Gerald and co-authors,
Overall, I enjoyed this manuscript. As an Indigenous STEM/public health scholar from the pacific, there are many themes that resonate with a worldview here. The weave and flow of the notions of water are represented throughout the manuscript. As a reader, I felt (mostly) walked through the terrain while having the important landmarks pointed out.
The ideas, the relevance to other concepts in the paper, and the meaning seemed to get clearer in the later sections of the manuscript. I would highly recommend reviewing the first couple of pages and ensuring you make the purpose and the relevance of the knowledge in the manuscript really clear. I have highlighted several examples in the below comments.
Major issues/comments to be addressed
Can you clarify the link or relationship between the purpose of the paper and the key ideas? Specifically, how do the seven distinct pathways outlined in the EA model (and details covered in subsequent lines 96-106) align or support the approaches you describe in lines 39 – 50?
Lines 65-68, beginning the sentence with ‘unfortunately’ leads to a lack of clarity in the intended message. This sentence has the potential to explicitly establish your research aim or focus, reword this sentence to reflect the premise of the manuscript.
Referring to lines 115 – 117 – How do the relationships with nibi as the system, relate to understanding the social impacts of environmental damage?
What peer-reviewed evidence is there to support your claim at line 119?
Can you include references to specific laws and policies that are relevant to your focus for lines 126-128?
What peer-reviewed evidence is there to support your claim on lines 258-259?
In the sentence on line 266, you say the effect on the stress of the population… is well documented, then goes on to mention one publication. Are there more publications that support your claims that the effect of stress is well documented?
Sentence on line 386 is unclear, what is happening with the health outcomes?
In reference to your conclusion, what are the outcomes of the interlocking ideas you established in your introduction? You explained the idea of interlocking 1. environmental affordance (inequalities), 2, the perspective of wellness, and 3. An ecosystem – lines 42-47, so what does that do for understanding the impact of environmental degradation on the relationships, experience of chronic stress, and the mental health outcome of WIFN people?
Minor issues/comments to be addressed
Could you explain environmental degradation at line 32? This is an important concept in this paper and to the assumptions, you make as authors and providing a brief description would ensure that your readers are able to follow.
Line 40 – include an ‘s’ – ‘… present protective factors’
In lines 39-42, you cover how you will not focus on pollutants, would environmental degradation be more suitable, as you are discussing ideas, situations and phenomena that are broader and more complex than pollutants.
Could you include an illustration or visual aid to the idea of interlocking 1. environmental affordance (inequalities), 2, the perspective of wellness, and 3. An ecosystem – lines 42-47?
Rewrite or reorder the sentence in lines 58-60, you shift from verb forms (e.g., swim and fly) to noun forms (e.g., the plants and trees), select either verb or nouns for the description of the local Beings.
Perhaps instead of using ‘thing’ at lines 77 and 422, you would consider ‘phenomena’?
Breakdown the sentence in lines 72-75 into multiple sentences so it is easier to read.
Is there an illustration or imagery of the EA model which would help the readers to visualise the paths?
Line 98 correct ‘… coping behaviours that are available on at WIFN’ – remove either on or at.
The sentence in lines 157-159 would read more clearly if you included ‘life’ “….and introducing their fast prior to puberty.”
You provide a wonderful description of WIFN, is there a suitable image to include/associate with like 225-236 which illustrates the variety and complexity of these unseeded territories?
How does the example on lines 289-291 represent or link to the consideration of the social changes and impacts between community, ecosystem and spiritual self?
The message in the paragraph starting at line 292 is unclear. What is the link or relationship between all aspects of Self and the affordances of social support? I can assume the message you are communicating but it requires a reflection on the paragraph before and the paragraph 292-306.
Within path three (lines 318-335) can you consider including anxiety or distress that impacts social relationships? Are there examples of the environmental stressors (which are found in the WIFN) that are linked to HPA axis activation or neurohormonal/neurochemical changes?
Consider including the evidence for colonised intergenerational trauma in relation to chronic stress as discussed in the paragraph starting on line 359.
You suggest the need for more research on lines 367 to 370 – can you explain what that suggested research might look like?
In the paragraph starting at 372, can you include or link to an example when discussing the application of traditional skills, especially as you had already mentioned Elders supporting younger folk to fish, which includes a social support element?
The suggested research at lines 392-394, what would this suggested research look like? Would it be important for WIFN to lead and explore such an idea? What would be the potential outcomes?
Correct ‘degreed’ line 431
I recommend proceeding with the publication once the above issues and comments have been addressed.
Author Response
Thank you for your extensive and insightful comments. We have responded to them as follows.
Major issues/comments to be addressed
Can you clarify the link or relationship between the purpose of the paper and the key ideas? Specifically, how do the seven distinct pathways outlined in the EA model (and details covered in subsequent lines 96-106) align or support the approaches you describe in lines 39 – 50?
We have added “We use the EA model specifically the seven pathways of this model allow us to shift of focus away from individual mental health to community mental wellness by taking into consideration the going impacts of colonization. In this context, colonization is seen as a shift to a global-capitalist system which uses the traditional territories of the local Anishinaabek as a dumping ground for environmental pollutant. Additionally, this model allows us to consider relationships, and changes to them, in a systematic fashion”
Lines 65-68, beginning the sentence with ‘unfortunately’ leads to a lack of clarity in the intended message. This sentence has the potential to explicitly establish your research aim or focus, reword this sentence to reflect the premise of the manuscript.
We have changes these lines to “Changes to affordances results in changes in relationships through alterations in use patterns and the resulting social relationships and the conditions that support mental wellness. Environmental degradation increases chronic stress through these changes, as well as through stress over the impacts of the pollutants themselves.”
Referring to lines 115 – 117 – How do the relationships with nibi as the system, relate to understanding the social impacts of environmental damage?
We have altered this statement as follows “In doing so, we reframe the system within which we are assessing the health impacts associated with environmental damage to one where relationships with nibi are the system rather than the imposed economic systems brought by global-capitalist/colonial power. By shifting our focus to nibi as they system we are able to embed our arguments within an Anishinaabek relational ontology rather than using an external system and authorities as our reference point.”
What peer-reviewed evidence is there to support your claim at line 119?
We have altered this statement to “ Based on Bruno Latour’s reminder that the “social” is born when two actors engage, we argue that all societies, which consist of more than two beings, have a value system that is based on the concept of relationships” and cited Latour’s work.
Can you include references to specific laws and policies that are relevant to your focus for lines 126-128?
We have added these laws to our references.
What peer-reviewed evidence is there to support your claim on lines 258-259?
We have added a peer-reviewed source that references damage to air, land, and water resulting from actions at Chemical Valley.
In the sentence on line 266, you say the effect on the stress of the population… is well documented, then goes on to mention one publication. Are there more publications that support your claims that the effect of stress is well documented?
We initially cited a paper written with members of the Walpole Island First Nation. We have now added additional papers to support this sentence. The area around Chemical Valley is very well studied because of the pollution.
Sentence on line 386 is unclear, what is happening with the health outcomes?
We have rephrased this topic sentence to be “The increase in negative mental and physical health outcomes associated with the ongoing environmental degradation in the territories of WIFN are not necessary, they are negative externalities caused by the decisions which led to a cluster of petrochemical plants on this stretch of river.”
In reference to your conclusion, what are the outcomes of the interlocking ideas you established in your introduction? You explained the idea of interlocking 1. environmental affordance (inequalities), 2, the perspective of wellness, and 3. An ecosystem – lines 42-47, so what does that do for understanding the impact of environmental degradation on the relationships, experience of chronic stress, and the mental health outcome of WIFN people?
We have added the following to the beginning of the conclusion: “We identify several benefits to integrating the EA model, an Anishinaabek perspective on wellness, and an ecosystem services approach. The first is a health equities approach to understanding mental health outcomes. This is an important transition away from individual approaches to mental illness which locate the problem in single individuals. The health equities approach provides a rebuttal to a lingering, racist, belief that First Nations people are somehow flawed and that these flaws are the cause of their poor health outcomes. We also see an important step in decolonizing discourse that is associated with Indigenous health through an shifting of that discourse to a local First Nation’s perspective rather then using local knowledge as an secondary form of knowing. We have worked to centre this paper on Nibi and relationships rather then economic gain and resulting degradation. Finally, we tie our argument to our understanding that local ecosystems shape communities through the process of ancestors learning how to live a good life in relationship with what is provided. From these teachings a community learns how to be healthy and when the relationships are damaged there is a greater risk of the community experiencing harms.”
Minor issues/comments to be addressed
Could you explain environmental degradation at line 32? This is an important concept in this paper and to the assumptions, you make as authors and providing a brief description would ensure that your readers are able to follow.
Reviewer two raised the same concern. We have added a definition for this term.
Line 40 – include an ‘s’ – ‘… present protective factors’
corrected
In lines 39-42, you cover how you will not focus on pollutants, would environmental degradation be more suitable, as you are discussing ideas, situations and phenomena that are broader and more complex than pollutants.
Thank you for this suggestion. We want to let readers know early that we are not actively measuring pollutant levels for this paper.
Could you include an illustration or visual aid to the idea of interlocking 1. environmental affordance (inequalities), 2, the perspective of wellness, and 3. An ecosystem – lines 42-47?
I’m not sure if our team has that skill set.
Rewrite or reorder the sentence in lines 58-60, you shift from verb forms (e.g., swim and fly) to noun forms (e.g., the plants and trees), select either verb or nouns for the description of the local Beings.
We have corrected this to a verb-based approach.
Perhaps instead of using ‘thing’ at lines 77 and 422, you would consider ‘phenomena’?
We have left “thing” in place in line 77 as that is the language used by the developers of the EA model. We have used your suggestion on line 422.
Breakdown the sentence in lines 72-75 into multiple sentences so it is easier to read.
We have reworded lives 72-75 to place greater emphasis on what the EA is and removed what it is not.
Is there an illustration or imagery of the EA model which would help the readers to visualise the paths?
We have inserted an illustration of the EA model
Line 98 correct ‘… coping behaviours that are available on at WIFN’ – remove either on or at.
Updated
The sentence in lines 157-159 would read more clearly if you included ‘life’ “….and introducing their fast prior to puberty.”
We have used “fast” in this context to denote a rapid pace of growth and development, rather than the important ceremonial fasting that traditionally happens prior to the onset of puberty.
You provide a wonderful description of WIFN, is there a suitable image to include/associate with like 225-236 which illustrates the variety and complexity of these unseeded territories?
Thank you. We will check with the journal to see if this is possible.
How does the example on lines 289-291 represent or link to the consideration of the social changes and impacts between community, ecosystem and spiritual self?
We have rephased this section to make it more about the loss of a relationship impacting the whole self.
The message in the paragraph starting at line 292 is unclear. What is the link or relationship between all aspects of Self and the affordances of social support? I can assume the message you are communicating but it requires a reflection on the paragraph before and the paragraph 292-306.
We have added this summary to help clarify this paragraph “As such, social supports are actualized through a performance of activities with our relations and when those actions are not available, because the affordances offered by the ecosystem are at risk, then mental health can be at risk.”
Within path three (lines 318-335) can you consider including anxiety or distress that impacts social relationships? Are there examples of the environmental stressors (which are found in the WIFN) that are linked to HPA axis activation or neurohormonal/neurochemical changes?
To our knowledge the only work that has been completed on biomarkers of stress in relation to environmental degradation at WIFN is the cortisol work. As a team we are interested in specific HPA axis research but have not been able to build a suitable team to secure the funding to this point in time.
Consider including the evidence for colonised intergenerational trauma in relation to chronic stress as discussed in the paragraph starting on line 359.
We have adapted this section to connect the intergenerational to a larger national conversation and cited a recent work on the topic.
You suggest the need for more research on lines 367 to 370 – can you explain what that suggested research might look like?
We have added “Specifically, we would like to understand cultural and biological mechanisms which connect perception through action so that we have a better way of understanding why risk causes actions.”
In the paragraph starting at 372, can you include or link to an example when discussing the application of traditional skills, especially as you had already mentioned Elders supporting younger folk to fish, which includes a social support element?
We have quoted and cited a recent study which found improved social connections in youth who were partnered with Elders to learn traditional skills.
The suggested research at lines 392-394, what would this suggested research look like? Would it be important for WIFN to lead and explore such an idea? What would be the potential outcomes?
WIFN always takes the lead in our research partnerships. We have updated this line to reflect the need for FN leadership in research.
Correct ‘degreed’ line 431
corrected
Major issues/comments to be addressed
Can you clarify the link or relationship between the purpose of the paper and the key ideas? Specifically, how do the seven distinct pathways outlined in the EA model (and details covered in subsequent lines 96-106) align or support the approaches you describe in lines 39 – 50?
We have added “We use the EA model specifically the seven pathways of this model allow us to shift of focus away from individual mental health to community mental wellness by taking into consideration the going impacts of colonization. In this context, colonization is seen as a shift to a global-capitalist system which uses the traditional territories of the local Anishinaabek as a dumping ground for environmental pollutant. Additionally, this model allows us to consider relationships, and changes to them, in a systematic fashion”
Lines 65-68, beginning the sentence with ‘unfortunately’ leads to a lack of clarity in the intended message. This sentence has the potential to explicitly establish your research aim or focus, reword this sentence to reflect the premise of the manuscript.
We have changes these lines to “Changes to affordances results in changes in relationships through alterations in use patterns and the resulting social relationships and the conditions that support mental wellness. Environmental degradation increases chronic stress through these changes, as well as through stress over the impacts of the pollutants themselves.”
Referring to lines 115 – 117 – How do the relationships with nibi as the system, relate to understanding the social impacts of environmental damage?
We have altered this statement as follows “In doing so, we reframe the system within which we are assessing the health impacts associated with environmental damage to one where relationships with nibi are the system rather than the imposed economic systems brought by global-capitalist/colonial power. By shifting our focus to nibi as they system we are able to embed our arguments within an Anishinaabek relational ontology rather than using an external system and authorities as our reference point.”
What peer-reviewed evidence is there to support your claim at line 119?
We have altered this statement to “ Based on Bruno Latour’s reminder that the “social” is born when two actors engage, we argue that all societies, which consist of more than two beings, have a value system that is based on the concept of relationships” and cited Latour’s work.
Can you include references to specific laws and policies that are relevant to your focus for lines 126-128?
We have added these laws to our references.
What peer-reviewed evidence is there to support your claim on lines 258-259?
We have added a peer-reviewed source that references damage to air, land, and water resulting from actions at Chemical Valley.
In the sentence on line 266, you say the effect on the stress of the population… is well documented, then goes on to mention one publication. Are there more publications that support your claims that the effect of stress is well documented?
We initially cited a paper written with members of the Walpole Island First Nation. We have now added additional papers to support this sentence. The area around Chemical Valley is very well studied because of the pollution.
Sentence on line 386 is unclear, what is happening with the health outcomes?
We have rephrased this topic sentence to be “The increase in negative mental and physical health outcomes associated with the ongoing environmental degradation in the territories of WIFN are not necessary, they are negative externalities caused by the decisions which led to a cluster of petrochemical plants on this stretch of river.”
In reference to your conclusion, what are the outcomes of the interlocking ideas you established in your introduction? You explained the idea of interlocking 1. environmental affordance (inequalities), 2, the perspective of wellness, and 3. An ecosystem – lines 42-47, so what does that do for understanding the impact of environmental degradation on the relationships, experience of chronic stress, and the mental health outcome of WIFN people?
We have added the following to the beginning of the conclusion: “We identify several benefits to integrating the EA model, an Anishinaabek perspective on wellness, and an ecosystem services approach. The first is a health equities approach to understanding mental health outcomes. This is an important transition away from individual approaches to mental illness which locate the problem in single individuals. The health equities approach provides a rebuttal to a lingering, racist, belief that First Nations people are somehow flawed and that these flaws are the cause of their poor health outcomes. We also see an important step in decolonizing discourse that is associated with Indigenous health through an shifting of that discourse to a local First Nation’s perspective rather then using local knowledge as an secondary form of knowing. We have worked to centre this paper on Nibi and relationships rather then economic gain and resulting degradation. Finally, we tie our argument to our understanding that local ecosystems shape communities through the process of ancestors learning how to live a good life in relationship with what is provided. From these teachings a community learns how to be healthy and when the relationships are damaged there is a greater risk of the community experiencing harms.”
Minor issues/comments to be addressed
Could you explain environmental degradation at line 32? This is an important concept in this paper and to the assumptions, you make as authors and providing a brief description would ensure that your readers are able to follow.
Reviewer two raised the same concern. We have added a definition for this term.
Line 40 – include an ‘s’ – ‘… present protective factors’
corrected
In lines 39-42, you cover how you will not focus on pollutants, would environmental degradation be more suitable, as you are discussing ideas, situations and phenomena that are broader and more complex than pollutants.
Thank you for this suggestion. We want to let readers know early that we are not actively measuring pollutant levels for this paper.
Could you include an illustration or visual aid to the idea of interlocking 1. environmental affordance (inequalities), 2, the perspective of wellness, and 3. An ecosystem – lines 42-47?
I’m not sure if our team has that skill set.
Rewrite or reorder the sentence in lines 58-60, you shift from verb forms (e.g., swim and fly) to noun forms (e.g., the plants and trees), select either verb or nouns for the description of the local Beings.
We have corrected this to a verb-based approach.
Perhaps instead of using ‘thing’ at lines 77 and 422, you would consider ‘phenomena’?
We have left “thing” in place in line 77 as that is the language used by the developers of the EA model. We have used your suggestion on line 422.
Breakdown the sentence in lines 72-75 into multiple sentences so it is easier to read.
We have reworded lives 72-75 to place greater emphasis on what the EA is and removed what it is not.
Is there an illustration or imagery of the EA model which would help the readers to visualise the paths?
We have inserted an illustration of the EA model
Line 98 correct ‘… coping behaviours that are available on at WIFN’ – remove either on or at.
Updated
The sentence in lines 157-159 would read more clearly if you included ‘life’ “….and introducing their fast prior to puberty.”
We have used “fast” in this context to denote a rapid pace of growth and development, rather than the important ceremonial fasting that traditionally happens prior to the onset of puberty.
You provide a wonderful description of WIFN, is there a suitable image to include/associate with like 225-236 which illustrates the variety and complexity of these unseeded territories?
Thank you. We will check with the journal to see if this is possible.
How does the example on lines 289-291 represent or link to the consideration of the social changes and impacts between community, ecosystem and spiritual self?
We have rephased this section to make it more about the loss of a relationship impacting the whole self.
The message in the paragraph starting at line 292 is unclear. What is the link or relationship between all aspects of Self and the affordances of social support? I can assume the message you are communicating but it requires a reflection on the paragraph before and the paragraph 292-306.
We have added this summary to help clarify this paragraph “As such, social supports are actualized through a performance of activities with our relations and when those actions are not available, because the affordances offered by the ecosystem are at risk, then mental health can be at risk.”
Within path three (lines 318-335) can you consider including anxiety or distress that impacts social relationships? Are there examples of the environmental stressors (which are found in the WIFN) that are linked to HPA axis activation or neurohormonal/neurochemical changes?
To our knowledge the only work that has been completed on biomarkers of stress in relation to environmental degradation at WIFN is the cortisol work. As a team we are interested in specific HPA axis research but have not been able to build a suitable team to secure the funding to this point in time.
Consider including the evidence for colonised intergenerational trauma in relation to chronic stress as discussed in the paragraph starting on line 359.
We have adapted this section to connect the intergenerational to a larger national conversation and cited a recent work on the topic.
You suggest the need for more research on lines 367 to 370 – can you explain what that suggested research might look like?
We have added “Specifically, we would like to understand cultural and biological mechanisms which connect perception through action so that we have a better way of understanding why risk causes actions.”
In the paragraph starting at 372, can you include or link to an example when discussing the application of traditional skills, especially as you had already mentioned Elders supporting younger folk to fish, which includes a social support element?
We have quoted and cited a recent study which found improved social connections in youth who were partnered with Elders to learn traditional skills.
The suggested research at lines 392-394, what would this suggested research look like? Would it be important for WIFN to lead and explore such an idea? What would be the potential outcomes?
WIFN always takes the lead in our research partnerships. We have updated this line to reflect the need for FN leadership in research.
Correct ‘degreed’ line 431
corrected
Major issues/comments to be addressed
Can you clarify the link or relationship between the purpose of the paper and the key ideas? Specifically, how do the seven distinct pathways outlined in the EA model (and details covered in subsequent lines 96-106) align or support the approaches you describe in lines 39 – 50?
We have added “We use the EA model specifically the seven pathways of this model allow us to shift of focus away from individual mental health to community mental wellness by taking into consideration the going impacts of colonization. In this context, colonization is seen as a shift to a global-capitalist system which uses the traditional territories of the local Anishinaabek as a dumping ground for environmental pollutant. Additionally, this model allows us to consider relationships, and changes to them, in a systematic fashion”
Lines 65-68, beginning the sentence with ‘unfortunately’ leads to a lack of clarity in the intended message. This sentence has the potential to explicitly establish your research aim or focus, reword this sentence to reflect the premise of the manuscript.
We have changes these lines to “Changes to affordances results in changes in relationships through alterations in use patterns and the resulting social relationships and the conditions that support mental wellness. Environmental degradation increases chronic stress through these changes, as well as through stress over the impacts of the pollutants themselves.”
Referring to lines 115 – 117 – How do the relationships with nibi as the system, relate to understanding the social impacts of environmental damage?
We have altered this statement as follows “In doing so, we reframe the system within which we are assessing the health impacts associated with environmental damage to one where relationships with nibi are the system rather than the imposed economic systems brought by global-capitalist/colonial power. By shifting our focus to nibi as they system we are able to embed our arguments within an Anishinaabek relational ontology rather than using an external system and authorities as our reference point.”
What peer-reviewed evidence is there to support your claim at line 119?
We have altered this statement to “ Based on Bruno Latour’s reminder that the “social” is born when two actors engage, we argue that all societies, which consist of more than two beings, have a value system that is based on the concept of relationships” and cited Latour’s work.
Can you include references to specific laws and policies that are relevant to your focus for lines 126-128?
We have added these laws to our references.
What peer-reviewed evidence is there to support your claim on lines 258-259?
We have added a peer-reviewed source that references damage to air, land, and water resulting from actions at Chemical Valley.
In the sentence on line 266, you say the effect on the stress of the population… is well documented, then goes on to mention one publication. Are there more publications that support your claims that the effect of stress is well documented?
We initially cited a paper written with members of the Walpole Island First Nation. We have now added additional papers to support this sentence. The area around Chemical Valley is very well studied because of the pollution.
Sentence on line 386 is unclear, what is happening with the health outcomes?
We have rephrased this topic sentence to be “The increase in negative mental and physical health outcomes associated with the ongoing environmental degradation in the territories of WIFN are not necessary, they are negative externalities caused by the decisions which led to a cluster of petrochemical plants on this stretch of river.”
In reference to your conclusion, what are the outcomes of the interlocking ideas you established in your introduction? You explained the idea of interlocking 1. environmental affordance (inequalities), 2, the perspective of wellness, and 3. An ecosystem – lines 42-47, so what does that do for understanding the impact of environmental degradation on the relationships, experience of chronic stress, and the mental health outcome of WIFN people?
We have added the following to the beginning of the conclusion: “We identify several benefits to integrating the EA model, an Anishinaabek perspective on wellness, and an ecosystem services approach. The first is a health equities approach to understanding mental health outcomes. This is an important transition away from individual approaches to mental illness which locate the problem in single individuals. The health equities approach provides a rebuttal to a lingering, racist, belief that First Nations people are somehow flawed and that these flaws are the cause of their poor health outcomes. We also see an important step in decolonizing discourse that is associated with Indigenous health through an shifting of that discourse to a local First Nation’s perspective rather then using local knowledge as an secondary form of knowing. We have worked to centre this paper on Nibi and relationships rather then economic gain and resulting degradation. Finally, we tie our argument to our understanding that local ecosystems shape communities through the process of ancestors learning how to live a good life in relationship with what is provided. From these teachings a community learns how to be healthy and when the relationships are damaged there is a greater risk of the community experiencing harms.”
Minor issues/comments to be addressed
Could you explain environmental degradation at line 32? This is an important concept in this paper and to the assumptions, you make as authors and providing a brief description would ensure that your readers are able to follow.
Reviewer two raised the same concern. We have added a definition for this term.
Line 40 – include an ‘s’ – ‘… present protective factors’
corrected
In lines 39-42, you cover how you will not focus on pollutants, would environmental degradation be more suitable, as you are discussing ideas, situations and phenomena that are broader and more complex than pollutants.
Thank you for this suggestion. We want to let readers know early that we are not actively measuring pollutant levels for this paper.
Could you include an illustration or visual aid to the idea of interlocking 1. environmental affordance (inequalities), 2, the perspective of wellness, and 3. An ecosystem – lines 42-47?
I’m not sure if our team has that skill set.
Rewrite or reorder the sentence in lines 58-60, you shift from verb forms (e.g., swim and fly) to noun forms (e.g., the plants and trees), select either verb or nouns for the description of the local Beings.
We have corrected this to a verb-based approach.
Perhaps instead of using ‘thing’ at lines 77 and 422, you would consider ‘phenomena’?
We have left “thing” in place in line 77 as that is the language used by the developers of the EA model. We have used your suggestion on line 422.
Breakdown the sentence in lines 72-75 into multiple sentences so it is easier to read.
We have reworded lives 72-75 to place greater emphasis on what the EA is and removed what it is not.
Is there an illustration or imagery of the EA model which would help the readers to visualise the paths?
We have inserted an illustration of the EA model
Line 98 correct ‘… coping behaviours that are available on at WIFN’ – remove either on or at.
Updated
The sentence in lines 157-159 would read more clearly if you included ‘life’ “….and introducing their fast prior to puberty.”
We have used “fast” in this context to denote a rapid pace of growth and development, rather than the important ceremonial fasting that traditionally happens prior to the onset of puberty.
You provide a wonderful description of WIFN, is there a suitable image to include/associate with like 225-236 which illustrates the variety and complexity of these unseeded territories?
Thank you. We will check with the journal to see if this is possible.
How does the example on lines 289-291 represent or link to the consideration of the social changes and impacts between community, ecosystem and spiritual self?
We have rephased this section to make it more about the loss of a relationship impacting the whole self.
The message in the paragraph starting at line 292 is unclear. What is the link or relationship between all aspects of Self and the affordances of social support? I can assume the message you are communicating but it requires a reflection on the paragraph before and the paragraph 292-306.
We have added this summary to help clarify this paragraph “As such, social supports are actualized through a performance of activities with our relations and when those actions are not available, because the affordances offered by the ecosystem are at risk, then mental health can be at risk.”
Within path three (lines 318-335) can you consider including anxiety or distress that impacts social relationships? Are there examples of the environmental stressors (which are found in the WIFN) that are linked to HPA axis activation or neurohormonal/neurochemical changes?
To our knowledge the only work that has been completed on biomarkers of stress in relation to environmental degradation at WIFN is the cortisol work. As a team we are interested in specific HPA axis research but have not been able to build a suitable team to secure the funding to this point in time.
Consider including the evidence for colonised intergenerational trauma in relation to chronic stress as discussed in the paragraph starting on line 359.
We have adapted this section to connect the intergenerational to a larger national conversation and cited a recent work on the topic.
You suggest the need for more research on lines 367 to 370 – can you explain what that suggested research might look like?
We have added “Specifically, we would like to understand cultural and biological mechanisms which connect perception through action so that we have a better way of understanding why risk causes actions.”
In the paragraph starting at 372, can you include or link to an example when discussing the application of traditional skills, especially as you had already mentioned Elders supporting younger folk to fish, which includes a social support element?
We have quoted and cited a recent study which found improved social connections in youth who were partnered with Elders to learn traditional skills.
The suggested research at lines 392-394, what would this suggested research look like? Would it be important for WIFN to lead and explore such an idea? What would be the potential outcomes?
WIFN always takes the lead in our research partnerships. We have updated this line to reflect the need for FN leadership in research.
Correct ‘degreed’ line 431
corrected
Major issues/comments to be addressed
Can you clarify the link or relationship between the purpose of the paper and the key ideas? Specifically, how do the seven distinct pathways outlined in the EA model (and details covered in subsequent lines 96-106) align or support the approaches you describe in lines 39 – 50?
We have added “We use the EA model specifically the seven pathways of this model allow us to shift of focus away from individual mental health to community mental wellness by taking into consideration the going impacts of colonization. In this context, colonization is seen as a shift to a global-capitalist system which uses the traditional territories of the local Anishinaabek as a dumping ground for environmental pollutant. Additionally, this model allows us to consider relationships, and changes to them, in a systematic fashion”
Lines 65-68, beginning the sentence with ‘unfortunately’ leads to a lack of clarity in the intended message. This sentence has the potential to explicitly establish your research aim or focus, reword this sentence to reflect the premise of the manuscript.
We have changes these lines to “Changes to affordances results in changes in relationships through alterations in use patterns and the resulting social relationships and the conditions that support mental wellness. Environmental degradation increases chronic stress through these changes, as well as through stress over the impacts of the pollutants themselves.”
Referring to lines 115 – 117 – How do the relationships with nibi as the system, relate to understanding the social impacts of environmental damage?
We have altered this statement as follows “In doing so, we reframe the system within which we are assessing the health impacts associated with environmental damage to one where relationships with nibi are the system rather than the imposed economic systems brought by global-capitalist/colonial power. By shifting our focus to nibi as they system we are able to embed our arguments within an Anishinaabek relational ontology rather than using an external system and authorities as our reference point.”
What peer-reviewed evidence is there to support your claim at line 119?
We have altered this statement to “ Based on Bruno Latour’s reminder that the “social” is born when two actors engage, we argue that all societies, which consist of more than two beings, have a value system that is based on the concept of relationships” and cited Latour’s work.
Can you include references to specific laws and policies that are relevant to your focus for lines 126-128?
We have added these laws to our references.
What peer-reviewed evidence is there to support your claim on lines 258-259?
We have added a peer-reviewed source that references damage to air, land, and water resulting from actions at Chemical Valley.
In the sentence on line 266, you say the effect on the stress of the population… is well documented, then goes on to mention one publication. Are there more publications that support your claims that the effect of stress is well documented?
We initially cited a paper written with members of the Walpole Island First Nation. We have now added additional papers to support this sentence. The area around Chemical Valley is very well studied because of the pollution.
Sentence on line 386 is unclear, what is happening with the health outcomes?
We have rephrased this topic sentence to be “The increase in negative mental and physical health outcomes associated with the ongoing environmental degradation in the territories of WIFN are not necessary, they are negative externalities caused by the decisions which led to a cluster of petrochemical plants on this stretch of river.”
In reference to your conclusion, what are the outcomes of the interlocking ideas you established in your introduction? You explained the idea of interlocking 1. environmental affordance (inequalities), 2, the perspective of wellness, and 3. An ecosystem – lines 42-47, so what does that do for understanding the impact of environmental degradation on the relationships, experience of chronic stress, and the mental health outcome of WIFN people?
We have added the following to the beginning of the conclusion: “We identify several benefits to integrating the EA model, an Anishinaabek perspective on wellness, and an ecosystem services approach. The first is a health equities approach to understanding mental health outcomes. This is an important transition away from individual approaches to mental illness which locate the problem in single individuals. The health equities approach provides a rebuttal to a lingering, racist, belief that First Nations people are somehow flawed and that these flaws are the cause of their poor health outcomes. We also see an important step in decolonizing discourse that is associated with Indigenous health through an shifting of that discourse to a local First Nation’s perspective rather then using local knowledge as an secondary form of knowing. We have worked to centre this paper on Nibi and relationships rather then economic gain and resulting degradation. Finally, we tie our argument to our understanding that local ecosystems shape communities through the process of ancestors learning how to live a good life in relationship with what is provided. From these teachings a community learns how to be healthy and when the relationships are damaged there is a greater risk of the community experiencing harms.”
Minor issues/comments to be addressed
Could you explain environmental degradation at line 32? This is an important concept in this paper and to the assumptions, you make as authors and providing a brief description would ensure that your readers are able to follow.
Reviewer two raised the same concern. We have added a definition for this term.
Line 40 – include an ‘s’ – ‘… present protective factors’
corrected
In lines 39-42, you cover how you will not focus on pollutants, would environmental degradation be more suitable, as you are discussing ideas, situations and phenomena that are broader and more complex than pollutants.
Thank you for this suggestion. We want to let readers know early that we are not actively measuring pollutant levels for this paper.
Could you include an illustration or visual aid to the idea of interlocking 1. environmental affordance (inequalities), 2, the perspective of wellness, and 3. An ecosystem – lines 42-47?
I’m not sure if our team has that skill set.
Rewrite or reorder the sentence in lines 58-60, you shift from verb forms (e.g., swim and fly) to noun forms (e.g., the plants and trees), select either verb or nouns for the description of the local Beings.
We have corrected this to a verb-based approach.
Perhaps instead of using ‘thing’ at lines 77 and 422, you would consider ‘phenomena’?
We have left “thing” in place in line 77 as that is the language used by the developers of the EA model. We have used your suggestion on line 422.
Breakdown the sentence in lines 72-75 into multiple sentences so it is easier to read.
We have reworded lives 72-75 to place greater emphasis on what the EA is and removed what it is not.
Is there an illustration or imagery of the EA model which would help the readers to visualise the paths?
We have inserted an illustration of the EA model
Line 98 correct ‘… coping behaviours that are available on at WIFN’ – remove either on or at.
Updated
The sentence in lines 157-159 would read more clearly if you included ‘life’ “….and introducing their fast prior to puberty.”
We have used “fast” in this context to denote a rapid pace of growth and development, rather than the important ceremonial fasting that traditionally happens prior to the onset of puberty.
You provide a wonderful description of WIFN, is there a suitable image to include/associate with like 225-236 which illustrates the variety and complexity of these unseeded territories?
Thank you. We will check with the journal to see if this is possible.
How does the example on lines 289-291 represent or link to the consideration of the social changes and impacts between community, ecosystem and spiritual self?
We have rephased this section to make it more about the loss of a relationship impacting the whole self.
The message in the paragraph starting at line 292 is unclear. What is the link or relationship between all aspects of Self and the affordances of social support? I can assume the message you are communicating but it requires a reflection on the paragraph before and the paragraph 292-306.
We have added this summary to help clarify this paragraph “As such, social supports are actualized through a performance of activities with our relations and when those actions are not available, because the affordances offered by the ecosystem are at risk, then mental health can be at risk.”
Within path three (lines 318-335) can you consider including anxiety or distress that impacts social relationships? Are there examples of the environmental stressors (which are found in the WIFN) that are linked to HPA axis activation or neurohormonal/neurochemical changes?
To our knowledge the only work that has been completed on biomarkers of stress in relation to environmental degradation at WIFN is the cortisol work. As a team we are interested in specific HPA axis research but have not been able to build a suitable team to secure the funding to this point in time.
Consider including the evidence for colonised intergenerational trauma in relation to chronic stress as discussed in the paragraph starting on line 359.
We have adapted this section to connect the intergenerational to a larger national conversation and cited a recent work on the topic.
You suggest the need for more research on lines 367 to 370 – can you explain what that suggested research might look like?
We have added “Specifically, we would like to understand cultural and biological mechanisms which connect perception through action so that we have a better way of understanding why risk causes actions.”
In the paragraph starting at 372, can you include or link to an example when discussing the application of traditional skills, especially as you had already mentioned Elders supporting younger folk to fish, which includes a social support element?
We have quoted and cited a recent study which found improved social connections in youth who were partnered with Elders to learn traditional skills.
The suggested research at lines 392-394, what would this suggested research look like? Would it be important for WIFN to lead and explore such an idea? What would be the potential outcomes?
WIFN always takes the lead in our research partnerships. We have updated this line to reflect the need for FN leadership in research.
Correct ‘degreed’ line 431
corrected
Major issues/comments to be addressed
Can you clarify the link or relationship between the purpose of the paper and the key ideas? Specifically, how do the seven distinct pathways outlined in the EA model (and details covered in subsequent lines 96-106) align or support the approaches you describe in lines 39 – 50?
We have added “We use the EA model specifically the seven pathways of this model allow us to shift of focus away from individual mental health to community mental wellness by taking into consideration the going impacts of colonization. In this context, colonization is seen as a shift to a global-capitalist system which uses the traditional territories of the local Anishinaabek as a dumping ground for environmental pollutant. Additionally, this model allows us to consider relationships, and changes to them, in a systematic fashion”
Lines 65-68, beginning the sentence with ‘unfortunately’ leads to a lack of clarity in the intended message. This sentence has the potential to explicitly establish your research aim or focus, reword this sentence to reflect the premise of the manuscript.
We have changes these lines to “Changes to affordances results in changes in relationships through alterations in use patterns and the resulting social relationships and the conditions that support mental wellness. Environmental degradation increases chronic stress through these changes, as well as through stress over the impacts of the pollutants themselves.”
Referring to lines 115 – 117 – How do the relationships with nibi as the system, relate to understanding the social impacts of environmental damage?
We have altered this statement as follows “In doing so, we reframe the system within which we are assessing the health impacts associated with environmental damage to one where relationships with nibi are the system rather than the imposed economic systems brought by global-capitalist/colonial power. By shifting our focus to nibi as they system we are able to embed our arguments within an Anishinaabek relational ontology rather than using an external system and authorities as our reference point.”
What peer-reviewed evidence is there to support your claim at line 119?
We have altered this statement to “ Based on Bruno Latour’s reminder that the “social” is born when two actors engage, we argue that all societies, which consist of more than two beings, have a value system that is based on the concept of relationships” and cited Latour’s work.
Can you include references to specific laws and policies that are relevant to your focus for lines 126-128?
We have added these laws to our references.
What peer-reviewed evidence is there to support your claim on lines 258-259?
We have added a peer-reviewed source that references damage to air, land, and water resulting from actions at Chemical Valley.
In the sentence on line 266, you say the effect on the stress of the population… is well documented, then goes on to mention one publication. Are there more publications that support your claims that the effect of stress is well documented?
We initially cited a paper written with members of the Walpole Island First Nation. We have now added additional papers to support this sentence. The area around Chemical Valley is very well studied because of the pollution.
Sentence on line 386 is unclear, what is happening with the health outcomes?
We have rephrased this topic sentence to be “The increase in negative mental and physical health outcomes associated with the ongoing environmental degradation in the territories of WIFN are not necessary, they are negative externalities caused by the decisions which led to a cluster of petrochemical plants on this stretch of river.”
In reference to your conclusion, what are the outcomes of the interlocking ideas you established in your introduction? You explained the idea of interlocking 1. environmental affordance (inequalities), 2, the perspective of wellness, and 3. An ecosystem – lines 42-47, so what does that do for understanding the impact of environmental degradation on the relationships, experience of chronic stress, and the mental health outcome of WIFN people?
We have added the following to the beginning of the conclusion: “We identify several benefits to integrating the EA model, an Anishinaabek perspective on wellness, and an ecosystem services approach. The first is a health equities approach to understanding mental health outcomes. This is an important transition away from individual approaches to mental illness which locate the problem in single individuals. The health equities approach provides a rebuttal to a lingering, racist, belief that First Nations people are somehow flawed and that these flaws are the cause of their poor health outcomes. We also see an important step in decolonizing discourse that is associated with Indigenous health through an shifting of that discourse to a local First Nation’s perspective rather then using local knowledge as an secondary form of knowing. We have worked to centre this paper on Nibi and relationships rather then economic gain and resulting degradation. Finally, we tie our argument to our understanding that local ecosystems shape communities through the process of ancestors learning how to live a good life in relationship with what is provided. From these teachings a community learns how to be healthy and when the relationships are damaged there is a greater risk of the community experiencing harms.”
Minor issues/comments to be addressed
Could you explain environmental degradation at line 32? This is an important concept in this paper and to the assumptions, you make as authors and providing a brief description would ensure that your readers are able to follow.
Reviewer two raised the same concern. We have added a definition for this term.
Line 40 – include an ‘s’ – ‘… present protective factors’
corrected
In lines 39-42, you cover how you will not focus on pollutants, would environmental degradation be more suitable, as you are discussing ideas, situations and phenomena that are broader and more complex than pollutants.
Thank you for this suggestion. We want to let readers know early that we are not actively measuring pollutant levels for this paper.
Could you include an illustration or visual aid to the idea of interlocking 1. environmental affordance (inequalities), 2, the perspective of wellness, and 3. An ecosystem – lines 42-47?
I’m not sure if our team has that skill set.
Rewrite or reorder the sentence in lines 58-60, you shift from verb forms (e.g., swim and fly) to noun forms (e.g., the plants and trees), select either verb or nouns for the description of the local Beings.
We have corrected this to a verb-based approach.
Perhaps instead of using ‘thing’ at lines 77 and 422, you would consider ‘phenomena’?
We have left “thing” in place in line 77 as that is the language used by the developers of the EA model. We have used your suggestion on line 422.
Breakdown the sentence in lines 72-75 into multiple sentences so it is easier to read.
We have reworded lives 72-75 to place greater emphasis on what the EA is and removed what it is not.
Is there an illustration or imagery of the EA model which would help the readers to visualise the paths?
We have inserted an illustration of the EA model
Line 98 correct ‘… coping behaviours that are available on at WIFN’ – remove either on or at.
Updated
The sentence in lines 157-159 would read more clearly if you included ‘life’ “….and introducing their fast prior to puberty.”
We have used “fast” in this context to denote a rapid pace of growth and development, rather than the important ceremonial fasting that traditionally happens prior to the onset of puberty.
You provide a wonderful description of WIFN, is there a suitable image to include/associate with like 225-236 which illustrates the variety and complexity of these unseeded territories?
Thank you. We will check with the journal to see if this is possible.
How does the example on lines 289-291 represent or link to the consideration of the social changes and impacts between community, ecosystem and spiritual self?
We have rephased this section to make it more about the loss of a relationship impacting the whole self.
The message in the paragraph starting at line 292 is unclear. What is the link or relationship between all aspects of Self and the affordances of social support? I can assume the message you are communicating but it requires a reflection on the paragraph before and the paragraph 292-306.
We have added this summary to help clarify this paragraph “As such, social supports are actualized through a performance of activities with our relations and when those actions are not available, because the affordances offered by the ecosystem are at risk, then mental health can be at risk.”
Within path three (lines 318-335) can you consider including anxiety or distress that impacts social relationships? Are there examples of the environmental stressors (which are found in the WIFN) that are linked to HPA axis activation or neurohormonal/neurochemical changes?
To our knowledge the only work that has been completed on biomarkers of stress in relation to environmental degradation at WIFN is the cortisol work. As a team we are interested in specific HPA axis research but have not been able to build a suitable team to secure the funding to this point in time.
Consider including the evidence for colonised intergenerational trauma in relation to chronic stress as discussed in the paragraph starting on line 359.
We have adapted this section to connect the intergenerational to a larger national conversation and cited a recent work on the topic.
You suggest the need for more research on lines 367 to 370 – can you explain what that suggested research might look like?
We have added “Specifically, we would like to understand cultural and biological mechanisms which connect perception through action so that we have a better way of understanding why risk causes actions.”
In the paragraph starting at 372, can you include or link to an example when discussing the application of traditional skills, especially as you had already mentioned Elders supporting younger folk to fish, which includes a social support element?
We have quoted and cited a recent study which found improved social connections in youth who were partnered with Elders to learn traditional skills.
The suggested research at lines 392-394, what would this suggested research look like? Would it be important for WIFN to lead and explore such an idea? What would be the potential outcomes?
WIFN always takes the lead in our research partnerships. We have updated this line to reflect the need for FN leadership in research.
Correct ‘degreed’ line 431
corrected
Reviewer 2 Report
This article describes the impact of environmental change and degradation on local mental health and wellness at Walpole Island First Nations. Authors contend that local knowledge that nibi (water) is essential to life should be central to efforts to maintain community mental health. Authors also implore a shift from human-centered ecosystems to bidirectional relationships that are protected against environmental degradation and risk to local mental health and wellness.
The review is as follows:
1. It would be good to know about more about the background and history of the Anishnaabek people. What are their demographics? Explain where Walpole Island First Nation (WIFN) is located. Lines 25-26 mention “the islands provided an abundance of foods, medicines, and knowledge to the ancestors of the current residents of the community”. What are examples of the foods, medicine, and knowledge? Describe this for the lay reader. This information appears to be in lines 225-240 but could be introduced earlier in the paper for context.
2. At the end of the Introduction, authors should state the objective of the paper. If the objective is already stated, it needs to be made clearer for the reader.
3. Before discussing the environmental degradation that impacts the relationships WIFN can have with each other and the ecosystem, it will be helpful to provide a definition of what constitutes environmental degradation.
4. There is an insightful and thoughtful discussion on the Anishinaabek Traditional Knowledge and Water and thought-provoking discussion provided on the comparison between the Western-Euro-Canadian and Anishinaabek value systems on the relationship between water and the environment (lines 119-142).
5. Lines 246-248 – In “The community has been forced to live with chemical spills from petro-chemical facilities in Sarnia’s “Chemical Valley,” which is located upstream on the St. Clair River”, explain when these spills took place. Are these ongoing occurrences?
6. In the Reference list, several of the references are duplicated more than twice (e.g., references 2-4, 5-6, 11 and 13-16, 25-27, 52-54). Please review.
Overall, this is a unique, interesting paper on an important and relevant topic. It is pleasing to see a paper on this topic. Expanding on some of the points can provide helpful insight to make the paper clearer. Review the reference list to check for duplication in citations.
Author Response
Thank you for taking the time to review our paper and making these insightful and important suggestions. Please see our responses below.
- It would be good to know about more about the background and history of the Anishnaabek people. What are their demographics? Explain where Walpole Island First Nation (WIFN) is located. Lines 25-26 mention “the islands provided an abundance of foods, medicines, and knowledge to the ancestors of the current residents of the community”. What are examples of the foods, medicine, and knowledge? Describe this for the lay reader. This information appears to be in lines 225-240 but could be introduced earlier in the paper for context.
We have added information on the Anishnaabek people in the introduction. We have added a short introduction on the foods and medicines to supplement the later discussion.
- At the end of the Introduction, authors should state the objective of the paper. If the objective is already stated, it needs to be made clearer for the reader.
We have added a clear articulat of the objective of the paper.
- Before discussing the environmental degradation that impacts the relationships WIFN can have with each other and the ecosystem, it will be helpful to provide a definition of what constitutes environmental degradation.
We have added a definition of environmental degradation in the introduction as you suggest.
- There is an insightful and thoughtful discussion on the Anishinaabek Traditional Knowledge and Water and thought-provoking discussion provided on the comparison between the Western-Euro-Canadian and Anishinaabek value systems on the relationship between water and the environment (lines 119-142).
Thank you.
- Lines 246-248 – In “The community has been forced to live with chemical spills from petro-chemical facilities in Sarnia’s “Chemical Valley,” which is located upstream on the St. Clair River”, explain when these spills took place. Are these ongoing occurrences?
We have added additional information to provide historical context.
- In the Reference list, several of the references are duplicated more than twice (e.g., references 2-4, 5-6, 11 and 13-16, 25-27, 52-54). Please review.
We have ensured that the reference list aligns with the journal’s expectations.
Round 2
Reviewer 2 Report
The authors have done well to address the suggested feedback. The revised paper is more detailed and comprehensive. I appreciate the additional background information on the Anishnaabek people to provide important context. It is also good to see the important discussion and emphasis on the continuing impacts of colonization and the use of the traditional territories of the local Anishinaabek as a dumping ground for environmental pollutants.
It is good to see a paper on this pertinent and compelling topic and the authors should be commended.
The only suggestion is for authors to do a proofread to catch some errors in spelling, punctuation, and related areas. Some examples from the Introduction include:
1) “We use the EA model specifically the seven pathways of this model allow us to”,
2) “taking into consideration the going impacts of colonization”,
3) “Our objective for this paper isto explore how”, and
4) “a global-capitalist system which uses the traditional territories of the local Anishinaabek as a dumping ground for environmental pollutant” (pollutant should be plural).
Once the language and style items are addressed, the paper appears suitable for publication.
Author Response
Thank you for your kind words and careful review of this paper. We have completed an additional proof read to remove the grammatical mistakes you pointed out plus a few others.